# Structural Analysis in Transit System Using Network Theory Case of Guadalajara, Mexico

**Orlando Barraza \* and Miquel Estrada** 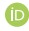

Civil Engineering School of Barcelona, Universitat Politècnica de Catalunya Technology, 08034 Barcelona, Spain; miquel.estrada@upc.edu

\* Correspondence: orlando.marath.barraza@upc.edu

**Abstract:** Structural analysis in a transit network is a key aspect used to evaluate in a planning process. In this sense, the use of network science was applied in this work to generate a framework of the main structural features of a transport network. In this case, an alternative transport network in Guadalajara, Mexico was taken as an example. The network properties selected were grade of accessibility, spatial friction, and vulnerability. In the case of the grade of accessibility, this propriety makes reference to the efficiency of the travel time that the network gives due to its structural features. The spatial friction measures how direct in terms of distance the trips that the network provides are, and the vulnerability relates to the ease with which the network can comprise its performance by affectations to their nodes or links. In this sense, this work presents a detailed methodology and a set of open-source tools that can be used to measure these key structural elements for decision making.

**Keywords:** transport networks; accessibility; network sciences; GTFS; data mining

## 1. Introduction

Humankind is involved in a civilizational crisis derived from human activity and an economic model that has been dominant up to now. The environmental degradation of natural resources and ecosystems coupled with climate change have the potential to destabilize entire regions. In this context, the rapid shifting to a sustainable development model is crucial in order to mitigate the potential crisis derived from the current human relationship with natural resources. Within the key changes in the parading of sustainable development, the change and improvement of critical infrastructure such as energy and transport are key elements in this transition. In the case of transport, there are multiples approaches with which to shift to a sustainable mobility system (understanding a sustainable mobility system as a one which minimizes the environmental impact, its energy demand, and that is competitive and economically viable). Some of these approaches include: Supporting infrastructure for non-motorized trips; land use plans which increase city density and improve the variety of land uses (15-min city); Increasing the competitiveness of public transport; Car management, to reduce its attractiveness (reducing parking spaces, zone restrictions, fuel prices, taxes, etc.); Acceleration of the electrification of the transport sector, and the migrating to a sharing economy paradigm of the transport sector.

Transit planning involves the integration of several fields of analysis, such as mobility dynamics, land use, and transit systems, to mention a few. In the case of the transit system, key conventional elements must be integrated into the analysis, such as the operation cost, emissions, energy demand, and general structure proprieties such as headway, the distance between stops, spatial coverage, the capacity of the lines, etc. [1,2].

However, there are some other key elements to analyze that help notoriously with the decision-making process and the evaluation of the performance of a transport network that is related to the physical and structural efficiency of the transport system. To measure these physical and structural features, transport planners have resorted to network sciences and the measure of network properties derived from this field. Network proprieties are rarely

taken into consideration in transport planning, despite the remarkable information that can be generate related to the performance of a transport network [3].

Network science is an academic field that uses discrete mathematics in order to understand the structural proprieties of a network. To do this, there is a necessity to have a network representation of the phenomena of study. This representation materializes in nodes and links that represent the dynamics and relations of a specific phenomenon [4]. Network science has become popular, especially since the 1990s, due to the improvement in computational power and the development of Geographical Information Systems (GIS) [5]. However, the difficulties in creating large datasets of networks have historically been a challenge in the field.

Nevertheless, the integration of complex transport networks is very recent [6]. In overall terms, these approaches have had a strong emphasis on routing optimization and transport costs [7]. To evaluate the proprieties of the transport system, a network representation must be done, this representation changes over time and over the kind of transport network subject to analysis. This characterization is generally a static representation; this allows the analysis of the structural or topological features of the transport system [8].

In the case of a transit system, most of the studies focus on a static approach, which tries to highlight the general features of the network in terms of topology, geometry, morphology, and traffic flows [6]. In terms of representation, transport networks can be represented as nodes and links where the nodes are the stops of the transport system and the links are the paths among each stop. This conception is the base of the representation of the transit system's configuration into a network, and is the baseline in order to analysis its structural features. Figure 1 shows this concept.

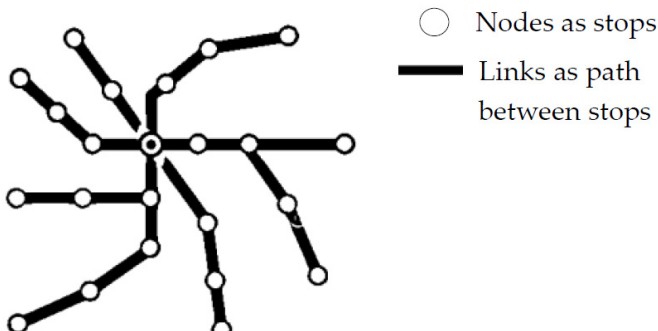

**Figure 1.** Example of the representation of a transport network.

Within this field of network science applied to transport networks, there are many network proprieties that can be measured in a transit system, such as the Diameter, Detour index, Network density, Pi index, Eta index, Theta index, Beta index, Alpha index, Gamma index, Koening number, Shimbel index, and Hub dependence, among others [9]. However, this paper measures three network properties, which are the Shimbel index, Detour index, and Hub dependence.

The Shimbel index is bond to the grade of accessibility which measures the structural efficiency in terms of travel time through the network [9]. On the other hand, the Detour index represents the spatial efficiency of the network, comparing the travel distance and the Euclidean distance among the origin and destination. Finally, the Hub dependence is a set of structural indicators bond with the resilience and vulnerability of the network [9]. The importance of measuring the network properties of a transit system relies on the premise that every main aspect of the performance of the transit system is bonded with its structure. Aspects such as operational cost, pollution, energy demand, and travel time, among others, are linked with the technology and structure of the transit network. This relation can be visualized in Figure 2 that represents the relationship that the structure and technology of the transit system have with other main aspects, such as operation cost, pollution, energy demand, and structural proprieties. As can be seen, the structure and technology of the

network are the main aspects that influence the performance of the rest of the relevant aspects of the transit system. If the transit system has efficient structure and technology, the operation cost is going to reduce, as well as the environmental impact and the energy demand. This influences network properties such as travel time, resiliency, and space friction. For this reason, transport planners must integrate the process of planning the network proprieties of the transit.

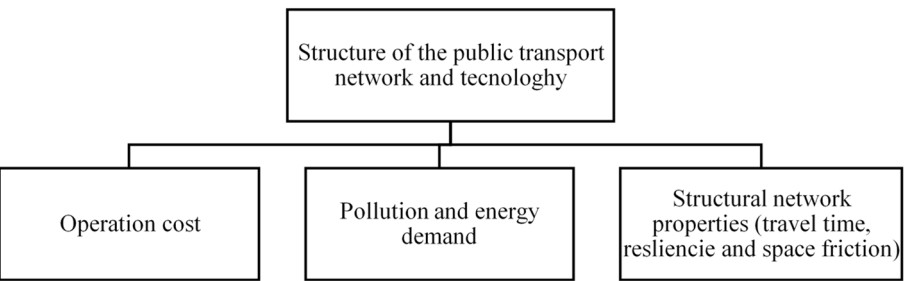

**Figure 2.** Conceptual relationship among main aspects of a transit system.

In this sense, the objective of this paper is to analysis some of the main structural features that should be taken into consideration in the planning process of a transits system and to generate a framework and a digital tool with which to carry out this analysis (accessibility, spatial friction, and vulnerability). This structural evaluation was taken into consideration as an alternative transit network proposed for Guadalajara, Mexico [10]. The network representation of this alternative transit system was made mainly using the general transit feed specification (GTFS). The GTFS is an accurate network representation, as it integrates the location of stops, the bond between stops in a route, and the travel time of the buses along each route [11]. To use and exploit the GTFS properly, a set of software tools were used. These tools mainly included the open route engine tool Open Trip Planner (OTP) and R as a programming platform.

Finally, this paper is divided into four sections: Section 1 gives a little introduction to the applications of network science in transit analysis, Section 2 addresses the methodology and the materials used, and Section 3 contains the results and discussion. Finally, Section 4 concludes of the work.

## 2. Materials and Methodology

### 2.1. Materials

In order to measure the network proprieties of the transit system, the following row data was used: zoning, centroids, GTFS, and street pattern. The zoning represents the area of interest within the Guadalajara Metropolitan Area (GMA); in total, the area of the GMA is around 628 km$^2$. However, the area of interest is near 330 km$^2$. Within the area of interest is 97% of the total population of the GMA, which reached 5 million inhabitants in 2020 [12]. Figure 3 shows the zoning and centroids created for the service area which consists of 398 hexagons with an apothem of 670 m.

On the other hand, Figure 4 shows the centroids of the area of interest. Each centroid represents the origin and destination for each trip where the network will provide the service.

Additionally, the street pattern data was obtained from Open Street Map. This data was used to integrated the walking distance of each trip. Finally, the last row data used to determine the network proprieties were the GTFS files for the alternative transit system. The GTFS of the alternative transit system had to be generated, and represents 2033 stops and 39 routes, with an orthogonal design (verticals and horizontal routes) that minimizes the use of routes, thus encouraging a transfer model network.

**Zoning of the area of interest**

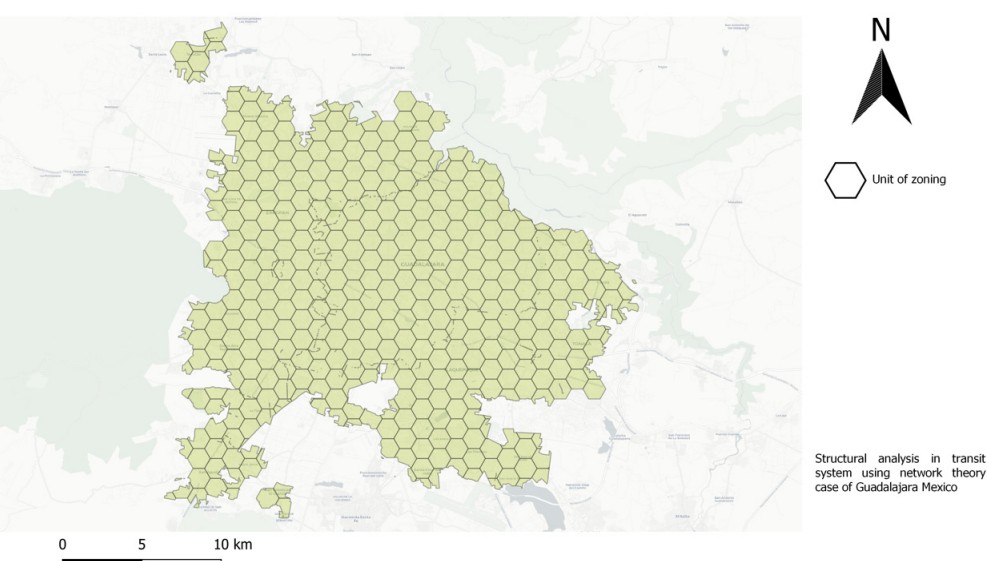

**Figure 3.** Zoning of the area of interest.

**Centroids of the area of interest**

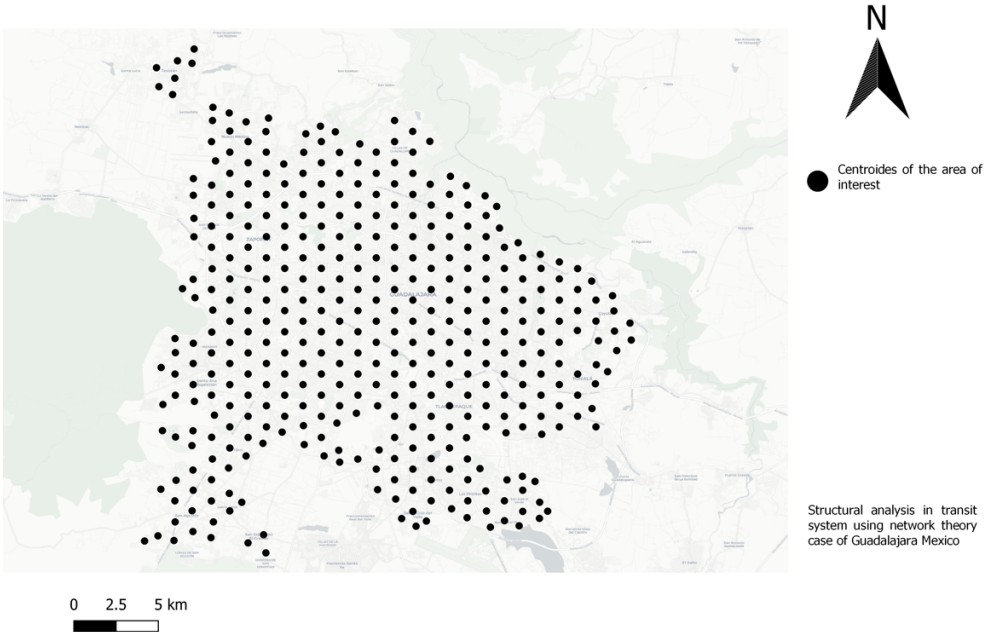

**Figure 4.** Centroids of the area of interest.

*2.2. Methodology*

The 3 network properties proposed for analysis in this work were: the Shimbel index (which represents the grade of accessibility and, hence, the travel time efficient), the Detour index (which represents the spatial friction), and the Hub dependence (which indicates the resilience and vulnerability of the network). Each of these network properties are described in the following points.

2.2.1. Shimbel Index and Grade of Accessibility

Shimbel index and the grade of accessibility express the efficiency in terms of travel time that the network gives, which can be interpreted as the travel time impedance [13].

In the case of the Shimbel index, it is a unique magnitude that expresses the efficiency in travel time in the whole network. In the case of the grade of accessibility, it can represent the efficiency in travel time at a nodal level. Both magnitudes are mainly influenced by the geometry of the transit network, its commercial speed, the headway of the lines, and the connections among the lines. Together, these aspects significantly determine the travel time efficiency. The following equations express both concepts:

$$C_{(x)} = \frac{N-1}{\sum_y d(x,y)} \tag{1}$$

where:

- $C_{(x)}$: The Shimbel index
- $d(x, y)$: Travel time between node $x$ and node $y$.
- $N$: Total number of nodes of the network

$$ATT_{(x)} = \frac{\sum_x d(x,y)}{N} \tag{2}$$

where:

- $ATT_{(x)}$ = Average travel time of the node $x$
- $\sum_x d(x,y)$ = The total sum of the travel time from node $x$ to the rest of the nodes represented by $y$
- $N$: Total nodes

### 2.2.2. Deuter Index

The Deuter index is a measure of the efficiency of a transport network in terms of distance. It can be understood as the spatial friction that the network gives. In this sense, the index takes 2 variables into consideration, which are the Euclidean distance between one pair of nodes and the real travel distance through the best route between a pair of nodes [9]. The following expression represents this relation:

$$DI = \frac{D(S)}{D(T)} \tag{3}$$

where:

- $DI$ = Deuter index
- $D(S)$: Euclidean distance between two points
- $D(T)$: Real distance using the network

As it is presented in Equation (2), the Deuter index, while closer to 1, is spatially more efficient as it reduces the friction of distance among the points. On a practical level, this means that the trips are more direct in spatial terms.

### 2.2.3. Hub Dependence of the Network

Hub dependence is the general term bond with the resilience and vulnerability that a network can have. Unlike the rest of the network indicators, there is no single network propriety that can be totally linked with the resilience and vulnerability of a network. For this reason, a set of network properties were proposed in this work to have a set of indicators relating to resilience and vulnerability [14]. These indicators are betweenness centrality, central point dominance, average path length, and spectral gap. Additionally, a set of scenarios were generated to see the affectation in terms of connectivity (measuring the average path length and the number of nodes connected using the Dijkstra algorithm). These scenarios were generated by removing random nodes and strategic nodes to see the impact that the network has.

The betweenness centrality measures the number of times a particular node is used as a bridge between the shortest path of two nodes. The measure can identify critical locations

of the network in case the network has a strong dependence on a few nodes (that is to say, that a few nodes are used extensively as part of the optimal path), meaning that there will be a problem of vulnerability. The performance of the total network could be compromised due to a disruption in this critical location of the city. The mathematical concept of the betweenness centrality is shown in the next equation.

$$C_i = \sum_{(j,k)} \frac{b_{jik}}{b_{jk}} \tag{4}$$

where:

- $Ci$ = Indicates the betweenness centrality of node $i$.
- $b_{jik}$: Indicates the total number of shortest paths between the node $j$ and $k$ that goes with using the node $i$.
- $b_{jk}$: Represents the total number of shortest paths between the node $j$ and $k$.

In the case of central point dominance, this network propriety consists of the average difference between the betweenness centrality of the most central point $B_{max}$ and all the other network nodes $Bi$ [14]. In simple terms, this means a measurement of node polarization of the networka lower value in the central point dominance means a more homogenous use of the nodes to make a trip, which, at the same time, means a more resilient network. This can be mathematically expressed in the following formula:

$$C_B = \frac{1}{n-1} \sum_1 (B_{max} - B_i) \tag{5}$$

where:

- $n$ = Number of nodes.
- $B_{max}$ = Maximum centrality.
- $B_i$ = Centrality of a given node.

On the other hand, the average path length is an indicator of the connectivity of the network. Its value indicates the average steps that the nodal has along the shortest paths to the rest of the nodes (it is the propriety of a network-level of aggregation, that is to say that the value of the average path length is for the entire network). In this sense, it measures the connectivity of the network. Additionally, the measurement of the average path length was performed in different scenarios of the removal nodes. One set of these scenarios was done by removing the random nodes of the network, changing the percentage of the nodal removal from 0% to 70%; the random nodal remotion was done in a cycle of 1000 times, generating a mean of the average path length of the 1000 repetitions. On the other hand, the other set of scenarios was performed by removing the critical nodes, which were the top nodes with the highest values of closeness centrality, and changing the value of remotion from 0% to 70%. This set of scenarios was done to see the affection of the connectivity of the network in cases of strategic nodal affectation in the center of the city. The following equation expresses the average path length:

$$l_g = \frac{1}{n(n-1)} \cdot \sum_{i \neq j} d(v_i, v_j) \tag{6}$$

where:

- $l_g$ = Average path length
- $n$ = Number of nodes
- $v_i, v_j$ = Shortest path among $v_i$ and $v_j$

Additionally, the spectral gap is defined as the difference among the moduli of the two largest eigenvalues of a matrix. The spectral gap provides information of the robustness of the network. It is used to detect networks with "good expansion" properties. A small

spectral gap would probably indicate the presence of articulations points or bridges that might cause serious disruptions of the flow in the network when removed [14]

$$\Delta\lambda \tag{7}$$

where:

- $\lambda$: eigenvalue

To get a clear idea behind the concept of spectral gap Figure 5 [15] shows an example of different configuration of networks and how it affects the value of the spectral gap and its bond with the robustness of the network. As it can be observed the spectral gap has a range value from 0 to 10. This is the real range of possible value of the spectral gap. In the Figure 5a there is an artificial network formed by two isolated components, which shows a spectral gap $\Delta\lambda = 0$. On the other hand, Figure 5b shows a network with a single connection among the two components which presence of a bottleneck. In this case the value of the spectral gap is 0.408. The inclusion of more links between the two components of the network increases the spectral gap for network in Figure 5c, which reach the maximum for the complete graph Figure 5d, where every pair of nodes is connected to each other [15].

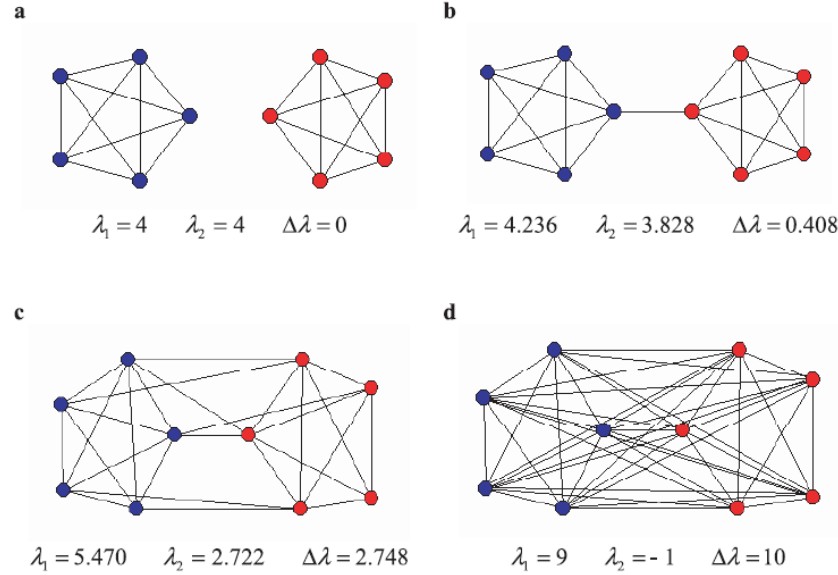

**Figure 5.** Spectral gap concept. (**a**), which shows a spectral gap $\Delta\lambda = 0$. The connection of these two components by a single link (**b**) represents an example of a network lacking good expansion properties due to the presence of a bottleneck. The spectral gap in this network is close to zero. The inclusion of more links between the two components of the network increases the spectral gap for network in (**c**), which reach the maximum for the complete graph (**d**), where every pair of nodes is connected to each other.

In the case of closeness centrality, this indicates a measurement of spatial centrality. It identifies the nodes that have a better location in the center of the network [16]. In this work, the measurement of closeness centrality was not integrated as a direct indicator of resilience. However, it was used to identify the nodes with the most centrality and use those nodes to remove strategic nodes from the network representing affectation in the center of the city. The following equation shows the mathematical expression of the closeness centrality and expresses the average path length:

$$C_c(i) = \frac{n-1}{\sum_{j=1}^{n} d(i,j)} \tag{8}$$

where

-     *Cc(i)*: Closeness centrality of the node
-     *n*: Total number of nodes
-     $\sum_{j=1}^{n} d(i,j)$: The sum of the shortest length among node *i* and the rest of the nodes

Moreover, to measure the number of nodes connected in a normal scenario, and in the set of scenarios of random nodal remotion and strategic nodal remotion, the Dijkstra algorithm was used. The Dijkstra algorithm aims to find the shortest path of one node to each of the rest of the nodes [17]. In this case, the algorithm was used just to identify the number of nodes connected.

### 2.2.4. Generation of Row Data with OTP

To generate the raw data necessary to measure the network properties (specifically, the Shimbel index and the Deuter index), the use of OTP and R was necessary. The reason to use OTP and R was for the generation of an N × N matrix of all the centroids of the area of service, and to generate a global data frame which contains the raw data (travel time and travel distance) used to generate the network proprieties of the Shimbel index and the Deuter index [18,19]. The steps followed for the generation of this row data can be summarized in the next points:

-     Selection of the area of study;
-     Creating zoning and centroids;
-     Set up of OTP in R, including data of street network and GTFS;
-     Generation of an N × N matrix among all the centroids;
-     Clean and process the data frame;
-     Generate Shimbel index and Deuter index.

The points presented above summarize the process used to generate the data frame that contains most of the row data used to generate the Shimbel index and the Deuter index. Figure 6 shows an example of the graphical of OTP.

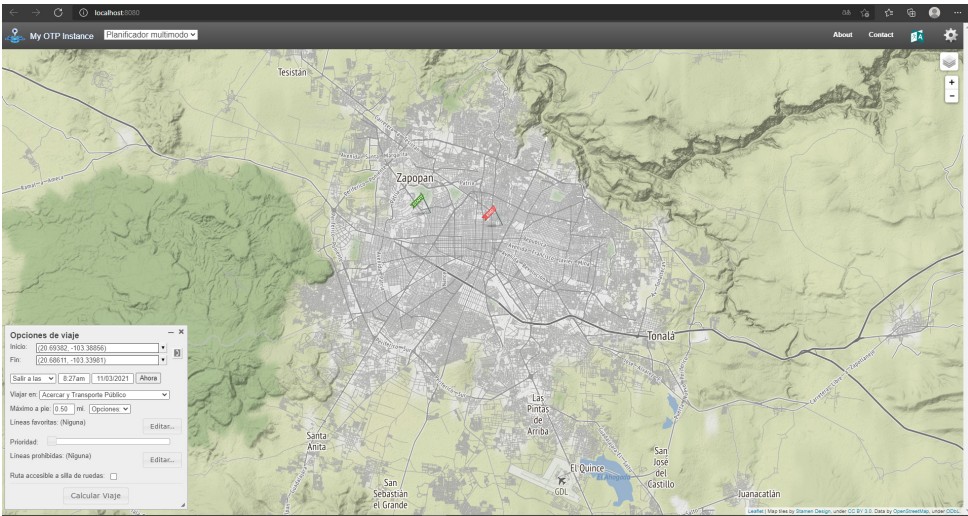

**Figure 6.** Example of the graphical view of OTP.

With the set-up of OTP in R, it is possible to generate a massive N × N matrix wherein every trip request generates the data presented in Table 1 which summarizes the main data generated by a trip request in OTP. However, extended data processing must be carried out in order to have the final result of the Shimbel index and the Deuter index.

**Table 1.** Data generated by a request in OTP.

| Duration | Start Time | End Time | Walk Time | Transit Time | Waiting Time | Walk Distance |
|---|---|---|---|---|---|---|
| 2680 s | 14 January 2021 8:24 | 14 January 2021 9:08 | 368 s | 2132 s | 180 s | 400 m |

| Transit Distance | Euclidian Distance | Transfers | Route | From Place | To Place | |
|---|---|---|---|---|---|---|
| 12,436 m | 9142 m | 0 | H4 | 651 | 987 | |

2.2.5. GTFS Transformation into Network Format

To generate the properties bond with the hub dependence (betweenness centrality, central point dominance, and average path length), it is necessary to convert the GTFS into a total graph format (specifically, to the format igraph used in *R* to analyze network proprieties). To convert the GTFS, a previous work was taken as a reference [20]. The main steps of this process are summarized in the next points:

- Lecture of the GTFS;
- Bind GTFS files;
- Join near stops within a given distance;
- Update coordinates of the new clusters of stations;
- Identify transport modes and routes for each trip;
- Identify links between stops;
- Calculate travel time between stops;
- Remove stops without connections;
- Build igrpah object.

The process presented above transforms the GTFS into an igraph object which is subject to fundamental network propriety measures.

**3. Results**

*3.1. Shimbel Index and Grade of Accessibility*

As was mentioned before, the Shimbel index and the grade of accessibility in a network indicates the travel time efficiency that the transit system gives. To have a spatial visualization of this propriety, the Figure 7 shows the grade of accessibility of each of the node (represented by a hexagon) each node contains a certain value of the average travel time that ranges from 45 min to 115 min.

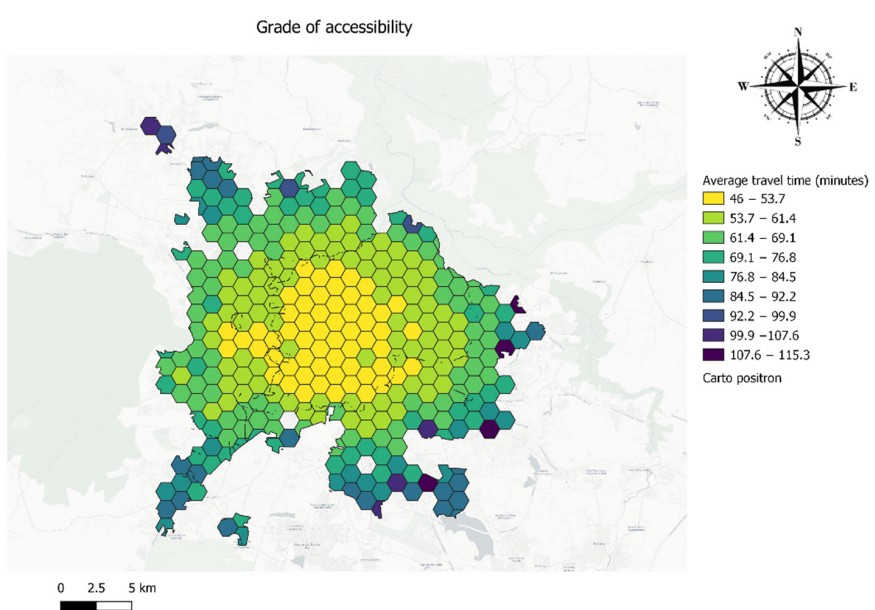

**Figure 7.** Grade of accessibility.

The average travel time is affected mainly by two factors. The first factor is due to its spatial location (the centroids that are more central have a lower average travel time), and the second factor is the features of the transport network. The features of the transport network, such as the geometry of the lines, the commercial speed, the headway, and the connectivity of the lines, have a significant impact on the efficiency of the reported travel time.

As it can be seen, the average travel time ranges from 46 min in some central areas of the city to 115 min in the periphery of the city. Figure 8 shows a box plot of the total travel time of every OD pair within the service area. 50% of the average travel time of all the centroids ranges from around 55 min to 70 min. Additionally, it can be observed that 75% of the trips have a travel time under 70 min.

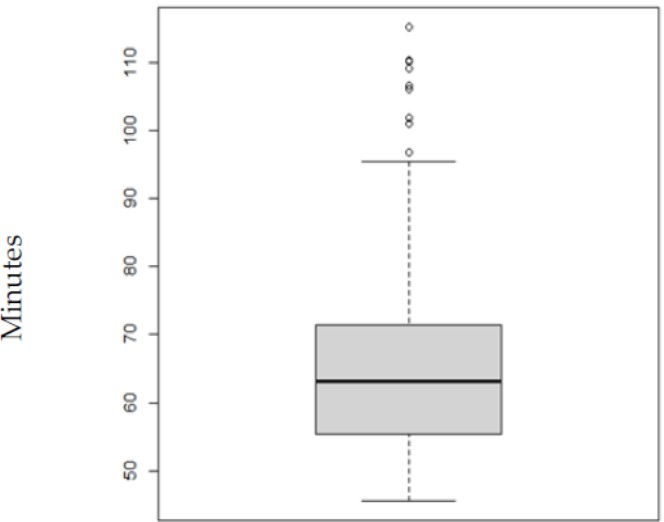

**Figure 8.** Box plot of the average travel time of every OD pair.

Finally, Table 2 contains the Shimbel index of the hole network, as well as the breakdown of average travel time. In the case of the average travel time, the value is 70 min composed by 52 min in vehicle travel time, 12 min of walking time and 6 min of waiting time. On the other hand, the Shimbel index has a value of 0.015. It is important to mention that, in order to compare the Shimbel index with another network, they have to contain the same number of nodes. If this does not happen, the Shimbel index is not compatible, and therefore able, to compare directly.

**Table 2.** Shimbel index and the average travel time.

| Parameter | Magnitude | Units |
|---|---|---|
| Shimbel index | 0.015 | Dimensionless |
| Average travel time | 70 | Minutes |
| Vehicle travel time | 52 | Minutes |
| Walking time | 12 | Minutes |
| Waiting time | 6 | Minutes |

*3.2. Deuter Index and Distance Friction*

In the case of the Deuter index, after the data processing, it was possible to generate a table that contains the distance of every trip using the transit system and the Euclidian distance for every origin and destination. Figure 9 shows a box plot of the distribution of the travel distance using the transit system and the Euclidian distance. The left side indicates the travel distance distribution on the transit system. As it can be seen, 75% of the trips are under 25 km. On the other hand, the Euclidian travel distance distribution has a travel distance of less than 15 km in the 75% of the trips.

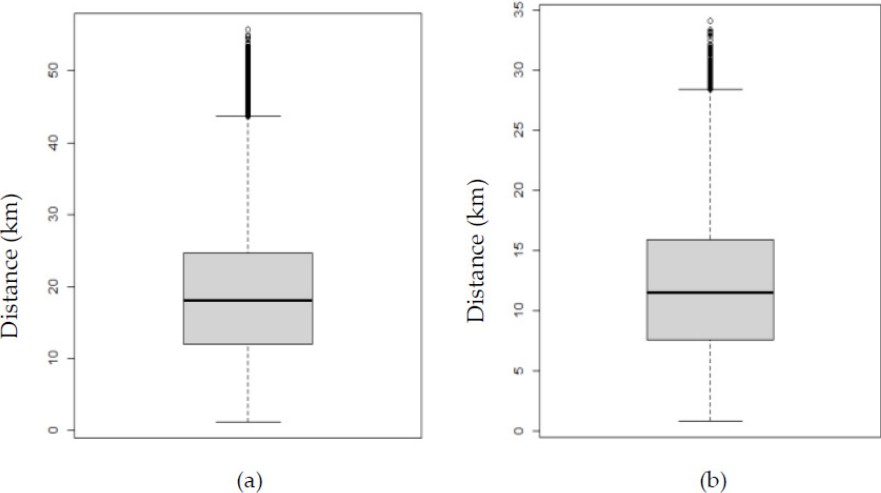

**Figure 9.** (**a**) represents a box plot of the transit travel distance distribution; (**b**) represents a box plot of the Euclidian travel distance distribution.

Additionally, Table 3 shows the final result of the Deuter index, the average travel distance using the transit system, and the average Euclidean distance. In the case of the Deuter index, it has a value of 0.63 (higher values means trips more direct in spatial terms), which was generated by dividing the Euclidian travel distance of 11.91 km and the regular travel distance using the transit system with a value of 18.81 km.

**Table 3.** Examples of the date generate calculating Deuter index.

| Parameter | Magnitude | Units |
|---|---|---|
| Deuter index | 0.63 | Dimensionless |
| Average travel distance using the transit network | 18.81 | km |
| Average Euclidean distance | 11.91 | km |

### 3.3. Hub Dependence and Resilience of the Network

The transformation of the GTFS into a graph object allowed us to measure the set of resilience and vulnerability indicators expressed previously. These indicators are betweenness centrality, central point dominance, and average path length. In the case of the set of indicators proposed for this section, the nodes are represented as a stop or a cluster of stops, and the links remain as the path that follows a route among each stop. Table 4 shows the results for the set of indicators. As can be observed, there are three measures of betweenness centrality, which are the highest, the average, and the lowest. The highest value reaches 746,551, which is the number of times a specific node was used as a bridge among the optimal path of an OD pair. The average value is of the betweenness centrality, and was 65,824. Meanwhile, the lowest was 0. These is a significant difference between each of the indicators. On the other hand, the average path length in typical conditions is 36.27. Finally, the spectral gap has a value of 0 which is a very low number that indicates a low level of robustness in the network.

Additionally, Figure 10 shows the boxplot of the distribution of the value of the betweenness centrality of all the nodes. As can be observed, 75% of the values in the betweenness centrality distribution are under around 75,000 units. The compact shape of the box indicates a highly similar distribution in the values of the betweenness centrality. However, as can be seen, the box plot has long whiskers that represent outliers in the value distribution. This means a much higher centrality in a few nodes compared with the rest of the nodes.

**Table 4.** Examples of the data generated for the betweenness centrality.

| Parameter | Magnitude | Units |
| --- | --- | --- |
| Highest betweenness centrality | 746,551 | dimensionless |
| Average betweenness centrality | 65,824 | km |
| Lowest betweenness centrality | 0 | dimensionless |
| Central point dominance | 680,726 | dimensionless |
| Average path length in normal condition | 36.27 | Stops |
| Spectral gap | 0.155 | dimensionless |

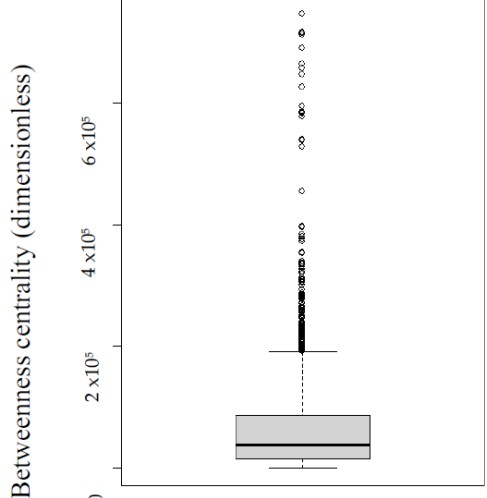

**Figure 10.** Boxplot of the betweenness centrality distribution.

On the other hand, the value of the betweenness centrality can be visualized spatially in Figure 11. The figure shows some key areas of the network. One of the most interesting is the segments of lines that cross the city horizontally in the south part of the service area. This specific area of the city is used as a bridge in so many trips. Other key areas can be identified in the center of the city and, as indicated by vertical line, in the western area. As a whole, these three areas are used very frequently as a bridge among the optimal trips within the service area.

The spatial visualization of the betweenness centrality allows the identification of key areas that can compromise the performance of the network (any perturbation in this area such as traffic, an accident, etc., can largely compromise the functionality of the transit system). On the other hand, the significant difference in the distribution value showed by the boxplot in Figure 10 indicates a great polarization of values.

In the case of the average path length, there were several scenarios of measurements. The first scenario of the average path length was performed in typical conditions (that is to say, all the nodes, stops, or areas of the city worked well without affecting the flow of trips among any stop), and has a value of 36.27 stops. On the other hand, different scenarios were done by randomly removing nodes from 0% to 70% of the nodal remotion (every one of these scenarios was measurement within a cycle of 1000 times in order to get the mean magnitude of the average path length in each scenario).

Finally, as was mentioned in Section 2, another set of scenarios were performed, removing strategic nodes (in this case, the nodes with the highest magnitude of closeness centrality and highest betweenness centrality results are presented in Figure 12. Figure 12 integrates three different scenarios. The first scenario is the average path length (connectivity) under random nodal remotion, the second scenarios is under strategic remotion

affecting the nodes with highest values in betweenness centrality and the third scenario is strategic nodal remotion for highest values in closeness centrality.

Betweenness centrality of the transit network

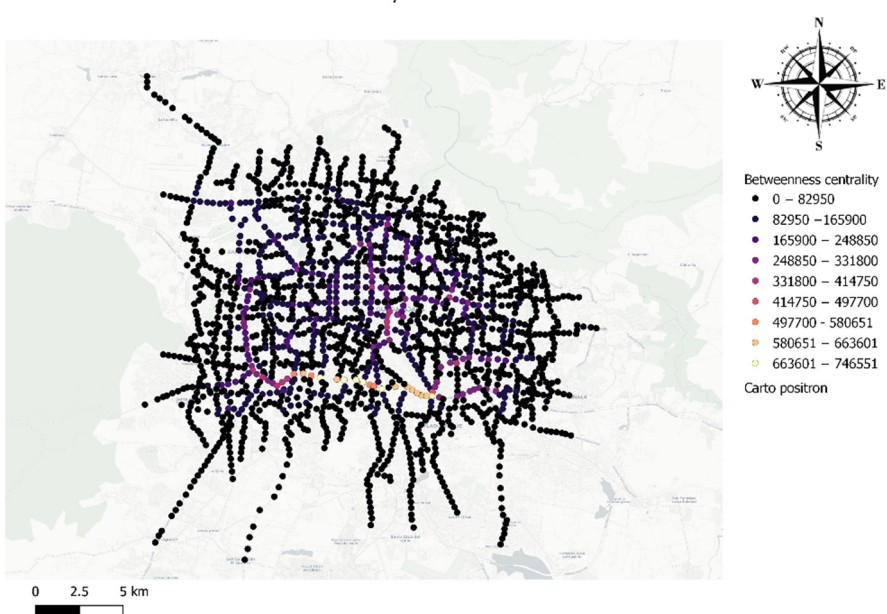

**Figure 11.** Betweenness centrality of the alternative network.

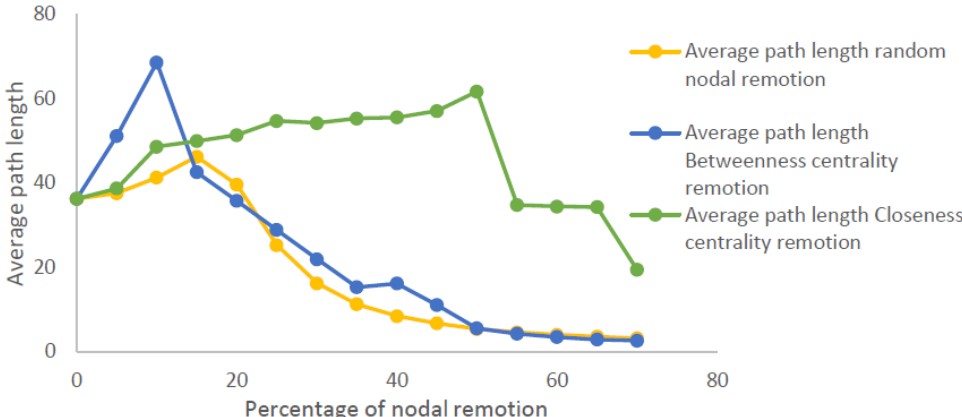

**Figure 12.** Average path length affectation in different scenarios of random nodal remotion.

The data used for built the Figure 12 can be seen in Table 5. As it can be observed in Table 5 the critical point under random remotion happened until the remotion of 15% of the nodes of the network in that point the connectivity passed from an average path length of 36 to 46 which is and affectation of 27%, this can be translated to travel time affectation which passed from an average of 70 to 89 min.

On the other hand, in the scenarios of strategic remotion using the nodes with highest betweenness centrality the critical point was reached at 10% of nodal remotion with a higher affectation passing from an average path length of 36 to 69 which means an affectation of 91%.

Finally, the critical point remotion the closeness centrality nodes was reached until the remotion of 50% of the nodes passing from an average path length of 36 to 62 which means an affectation of 72%. Under these scenarios can be visualized the most import affectation is done under scenarios of betweenness centrality remotion. This data showed the importance of identification the nodes with highest betweenness centrality, due to

these nodes are the most used within the shortest trips the affectation in these nodes have remarkable affectation in the connectivity of the network.

**Table 5.** Average path length and travel time affectation in different scenario of nodal remotion.

| Percentage of Nodal Remotion | Average Path Length under Different Scenarios | | | Travel Time Affectation (Minutes) | | |
|---|---|---|---|---|---|---|
| | Random Remotion Scenarios | Remotion of Top Betweenness Centrality | Remotion of Top Closeness Centrality | Random Remotion Scenario | Remotion of Top Betweenness Centrality | Remotion of Top Closeness Centrality |
| 0 | 36 | 36 | 36 | 70 | 70 | 70 |
| 5 | 37 | 51 | 39 | 73 | 99 | 75 |
| 10 | 41 | **69** | 49 | 80 | 133 | 94 |
| 15 | **46** | 43 | 50 | 89 | 82 | 97 |
| 20 | 39 | 36 | 51 | 77 | 69 | 99 |
| 25 | 25 | 29 | 55 | 49 | 56 | 106 |
| 30 | 16 | 22 | 54 | 31 | 42 | 105 |
| 35 | 11 | 15 | 55 | 22 | 30 | 107 |
| 40 | 8 | 16 | 56 | 16 | 31 | 107 |
| 45 | 6 | 11 | 57 | 13 | 21 | 110 |
| 50 | 5 | 6 | **62** | 10 | 11 | 119 |
| 55 | 4 | 4 | 35 | 9 | 8 | 67 |
| 60 | 4 | 3 | 34 | 8 | 7 | 67 |
| 65 | 3 | 3 | 34 | 7 | 6 | 66 |
| 70 | 3 | 3 | 19 | 6 | 5 | 37 |

In addition to this analysis, as was mentioned in Section 2, the measurement of the total nodes connected was conducted using the Dijkstra algorithm for the different scenarios. Figure 13 illustrates the total number of connected nodes. All the scenarios (random remotion, and both strategic remotion) present numbers of total connected nodes very similar. All of them start in 3.48 million of possible connection. In the case of the random remotion, it ends with 877,032 connected nodes. On the other end, the scenario of Betweenness centrality remotion ends with 884,540 nodes and 478,172 for the scenario of Closeness centrality. In general terms there is not a significant different in the total possible connection among all the scenarios. In this sense, the network has a notable difference in the performance of its connectivity under the different scenarios but all of them still are able to connected about the same number of nodes.

As it can be observed in this section, the variety of results of network proprieties indicators give a good overall view of the structural performance of the network in terms of travel time (grade of accessibility), spatial friction (Deuter index), and vulnerability.

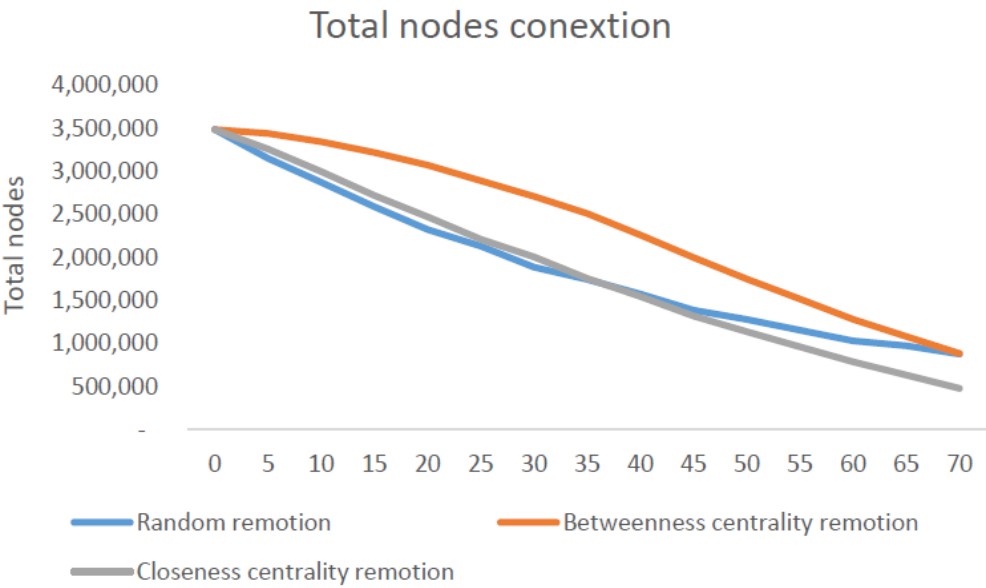

**Figure 13.** Total connected nodes under random nodal remotion and strategic remotion.

## 4. Conclusions

The methodology presented in this work is a robust framework, able to be applied to any transit network using the code developed for this work. The aspects for measurement proposed (accessibility, spatial friction, and vulnerability) are some of the main relevant aspects that a planner must take into consideration in order to compare the performance of different transit network scenarios within a planning process. In this sense, the work done here is a solid framework and a tool that can be used widely.

The results presented here show a clear panorama of the performance of the structural aspects of the transit system. In the case of accessibility, it showed a better performance in the center of the city. This follows a basic principle of centrality: the areas of the city in the center will have better connectivity to the rest of the area of interested due to its location. However, other variables get into account which are the commercial speed of the buses, the geometry of the lines, the headway and the connectivity among the routes. However, in order to take this as a real indicator for decision-making, a comparison between transit system or modes of transport should take place, in order to see understand the performance among different scenarios or modes of transport and select generate a decision towards the best scenario. In the case of the alternative network examined in this study, the average travel time was 70 min, and it showed a travel time distribution close to a normal distribution. In general terms, this distribution and the magnitude of the travel time are good indicators of the performance of the network in this aspect.

In the case of spatial friction, due to the fact that the alternative network was based on a grid layout, that is to say, a network based on transfers that allows the user go from and to anywhere with one transfer, the spatial friction was low. In this case, the Deuter index was 0.63. Finally, the vulnerability analysis was the most significant. In the case of the vulnerability analysis, it showed a poor performance with a great polarization to some of the nodes (stops) of the network.

This was very visible with the shape of the box plot in Figure 10 and with the central point dominance that had a value of 680,726 when the mean of the betweenness centrality was 65,824. This polarization paved the way for affectation in the connectivity. The impact in the connectivity was visible in Table 5. This analysis showed the key relevance that has the nodes with highest betweenness centrality to the connectivity of the network. The remotion in these nodes trigger the most remarkable impacts to the connectivity passing from an average of 70 min to 133 min in the critical point which was at 10% of nodal remotion. As a conclusion from this analysis in order to keep the performance of the

transit system transport planners must identify the key nodes of the network and generates policies dedicated to keep the optimal flow of the transit system in those zones. As it was seem in this work little perturbations on these nodes can led to deep impacts in the average travel time for users.

While it is true that the results are solid numbers for the alternative transit network used in this work, it is difficult to make a general judgment about the network's efficiency due to the fact that there is no other network with which to compare it to, as a consequence of a lack of data (GTFS) related the actual transit network within the service area. This lack of data made it impossible to do these full analyses by comparing two transit networks in the same area of service.

For this reason, it is hard to generate a final judgment of the performance of this alternative network. However, some elements of this work must be highlighted, such as the fact that an open-source tool in R was generated in order to have a solid framework with which to analyze the structural features of transit networks [21].

**Author Contributions:** Conceptualization, O.B.; methodology, O.B.; software, O.B.; validation, Miquel Estrada; formal analysis, O.B.; investigation, M.E. and O.B.; resources, O.B.; data curation, O.B.; writing—original draft preparation, O.B.; writing—review and editing, O.B.; visualization, O.B.; supervision, M.E.; project administration, M.E.; funding acquisition, O.B. and M.E. All authors have read and agreed to the published version of the manuscript.

**Funding:** This research was funded by Mexican National Council for Science and Technology (CONACYT), grant number 641618.

**Institutional Review Board Statement:** Not applicable.

**Informed Consent Statement:** Not applicable.

**Data Availability Statement:** Not applicable.

**Acknowledgments:** The authors are grateful with because the technical and financial support made by the Universitat Politècnica de Catalunya (UPC) and the Mexican National Council for Science and Technology (CONACYT).

**Conflicts of Interest:** The authors declare no conflict of interest.

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
