# Peer review of "Structural Analysis in Transit System Using Network Theory Case of Guadalajara, Mexico"

_urbansci, doi:10.3390/urbansci5040087_

Round 1

Reviewer 1 Report

General comments:

This paper is very interesting in the optical of SDGs.The methodology is presented into detail but the paper presentation could be improved.

The results obtained are highlighted.

Some revisions are necessary after the publication.

Introduction should be improved. The objectives of the work should be highlighted into detail in the paper.

The quality of the Figures and the presentation of the paper could be improved. 

Minor revisions:

-ABSTRACT:

"In this sense, this work presents a detailed methodology and set of open-source tools that can be used to measure these key structural elements for decision making". Please, check this character respect to the other text in the abstract.

INTRODUCTION:

Please add other References with recent publication years to underline the actuality of the research topic. The literature review could be enriched.

Please try to contextualize the problem in the introduction considering the general aim of the topic (Sustainable Development Goal, Human health…)

"the capacity of the lines, etc.".

Please here add this references:

-Carrese S., Cuneo, V., Nigro, M., Pizzuti, R., Ardito, C.F., Marseglia, G. Optimization of downstream fuel logistics based on road infrastructure conditions and exposure to accident events. Transport Policy. In Press, Reference, In Press, 2019. ISSN 0967-070X,https://doi.org/10.1016/j.tranpol.2019.10.016.

-Marseglia G., Medaglia, C.M., Ortega, F.A., Mesa, J.A., Optimal alignments for designing urban transport systems: application to Seville. Sustainability 2019, 11(18), 5058; https://doi.org/10.3390/su11185058.

Figure 1: Please try to improve the quality of Figure 1. Try to use different colours in this image.

 The introduction should be enriched:

-Please add some lines on the use of heuristic and meta heuristic approach in this kind of problem.

PLeas improve the quality of Figures 3 and 6.

Figure 11: Please specify the measure unit of y-axis(length).

x-axis: Percentage and not percetage..

Conclusions: Please in the conclusion paragraph underline the point of force of this paper and the importance of the obtained results.

References: you have to improve the number of references for your state of the art.

Author Response

This document contains the corrections of your coments.

Reviewer 2 Report

Refer to the attached file, please.

Author Response

This document contains the corrections of your comments. 

Round 2

Reviewer 2 Report

I think the manuscript was updated which is suitable for publication.